# MYC Causes Multiple Myeloma Progression via Attenuating TP53-Induced MicroRNA-34 Expression

**DOI:** 10.3390/genes14010100

**Published:** 2022-12-29

**Authors:** Yuki Murakami, Kei Kimura-Masuda, Tsukasa Oda, Ikuko Matsumura, Yuta Masuda, Rei Ishihara, Saki Watanabe, Yuko Kuroda, Tetsuhiro Kasamatsu, Nanami Gotoh, Hisashi Takei, Nobuhiko Kobayashi, Takayuki Saitoh, Hirokazu Murakami, Hiroshi Handa

**Affiliations:** 1Department of Laboratory Sciences, Gunma University Graduate School of Health Sciences, Maebashi 371-8510, Japan; 2Laboratory of Mucosal Ecosystem Design, The Institute for Molecular and Cellular Regulation, Gunma University, Maebashi 371-8510, Japan; 3Department of Hematology, Gunma University Graduate School of Medicine, Maebashi 371-8510, Japan; 4Faculty of Medical Technology and Clinical Engineering, Gunma University of Health and Welfare, Maebashi 371-0823, Japan

**Keywords:** multiple myeloma, epigenetics, microRNA, p53, MYC

## Abstract

MicroRNAs (miRNAs and miRs) are small (19–25 base pairs) non-coding RNAs with the ability to modulate gene expression. Previously, we showed that the miR-34 family is downregulated in multiple myeloma (MM) as the cancer progressed. In this study, we aimed to clarify the mechanism of miRNA dysregulation in MM. We focused particularly on the interaction between MYC and the TP53-miR34 axis because there is a discrepancy between increased TP53 and decreased miR-34 expressions in MM. Using the nutlin-3 or Tet-on systems, we caused wild-type (WT) p53 protein accumulation in human MM cell lines (HMCLs) and observed upregulated miR-34 expression. Next, we found that treatment with an Myc inhibitor alone did not affect miR-34 expression levels, but when it was coupled with p53 accumulation, miR-34 expression increased. In contrast, forced MYC activation by the MYC-ER system reduced nutlin-3-induced miR-34 expression. We also observed that TP53 and MYC were negatively correlated with mature miR-34 expressions in the plasma cells of patients with MM. Our results suggest that MYC participates in the suppression of p53-dependent miRNA expressions. Because miRNA expression suppresses tumors, its inhibition leads to MM development and malignant transformation.

## 1. Introduction

Cancer pathogenesis and progression are largely driven by mutations that cause abnormal gene expression and functional alterations [1,2]. However, epigenetic changes that regulate gene expression, including DNA promoter methylation and histone modification, also play important roles in oncogenesis [2,3]. We now know that non-coding RNAs are influential in epigenetics; despite previously being considered as ‘junk RNAs’ because they are not translated into proteins, many of these RNAs actually regulate gene expressions [4]. In particular, microRNAs (miRNAs)—small non-coding RNAs of 19–25 nucleotides—silence gene expressions by degrading target messenger RNA (mRNA) or by inhibiting its translation [4,5].

Multiple myeloma (MM) is a plasma cell malignancy [6,7]. Almost all patients with MM progress from a pre-malignant stage called monoclonal gammopathy of undetermined significance (MGUS). Chromosomal abnormalities are involved in the genesis of clonal plasma cells, and oncogene mutations, such as RAS mutations, play critical roles in disease progression [8]. However, all the steps in the progression from MGUS to MM have not been fully elucidated.

Epigenetic mechanisms are involved in MM pathogenesis and progression [9]. Numerous reports have described aberrant miRNA expression in MM [10,11,12]. Likewise, we previously showed that the miR-34, miR-29, miR-15a, and mi-R16 families are downregulated in this cancer [13,14]. However, the mechanisms underlying miRNA downregulation remain unclear. 

Members of the miR-34 family are tumor suppressors upregulated by TP53 [15]. Previous studies have shown that MYC downregulates several tumor-suppressive miRNAs, such as the miR-29 family [16]. Thus, we hypothesized that MYC acts on TP53 to suppress miR-34 upregulation in MM, and we attempted to elucidate this mechanism by using TP53-inducible MM cell line models.

## 2. Materials and Methods

### 2.1. Cell Lines

The cell lines used in this study were HMCLs MM.1S, KMS27, KMS28BM, KMS26, OPM2, and KMS11 and human bone osteosarcoma U2OS. MM.1S was obtained from the Deutsche Sammlung von Mikroorganismen und Zellkulturen (DSMZ, Braunschweig, Germany). The U2OS cells were purchased from the American Type Culture Collection (ATCC, Manassas, VA, USA). KMS27 and KMS28BM were provided by Dr. Hideto Tamura (Nippon Medical School, Tokyo, Japan). The KMS26 and KMS11 cells were provided by Dr. Takemi Otsuki (Kawasaki Medical School, Okayama, Japan). OPM2 was provided by Dr. Masaki Ri (Nagoya City University, Aichi, Japan). Human MCLs were cultured in RPMI-1640 medium (Sigma-Aldrich, St Louis, MO, USA), while U2OS cells were cultured in Eagle’s medium (DMEM; FUJIFILM Wako Pure Chemical Corporation, Osaka, Japan) supplemented with 10% fetal bovine serum at 37 °C in 5% CO_2_.

### 2.2. Patients

The study used bone marrow plasma cells from 109 patients with newly diagnosed MM and from 64 patients with MGUS. Plasma cells were purified from bone marrow mononuclear cells using anti-CD138 antibody conjugated with phycoerythrin (PE) (Beckman-Coulter, Brea, CA, USA) and an Easy Sep PE positive selection kit containing anti-PE antibody conjugated with micro-magnetic beads (STEMCELL Technologies, Vancouver, BC, Canada). The patients were diagnosed with MM or MGUS between July 2010 and March 2015. This study was approved by the institutional review board of Gunma University Hospital and followed all guidelines under the Declaration of Helsinki. Informed consent was obtained from all patients. The patients’ demographics are shown in Table 1.

### 2.3. Treatment with MDM2 Inhibitor Nutlin-3

The myeloma cell lines MM.1S, KMS27, KMS28BM, KMS26, OPM2, and KMS11 were treated with 1 or 10 µM (-)-Nutlin-3 (Cayman Chemical, Ann Arbor, MI, USA). Real-time PCR was used to determine gene expression after 48 h (MM.1S) or 72 h (remaining HMCLs) of treatment. Cell proliferation was determined using WST-8 assays (Dojindo Laboratories, Kumamoto, Japan).

### 2.4. p53 Overexpression Using Tet-On System

Doxycycline-inducible lentivirus vector pCW57.1 (plasmid #41393) and R777-E351 Hs.*TP53* (plasmid #70635) encoding WT *TP53* were purchased from Addgene (Cambridge, MA, USA). R777-E351 Hs.*TP53* was inserted into pCW57.1 using the Gateway^®^ LR Clonase™ Enzyme Mix (Thermo Fisher Scientific, Waltham, MA, USA). pCW57.1-*TP53* was amplified using a GenElute™ Plasmid Midiprep Kit (Sigma-Aldrich, St Louis, MO, USA). Pseudotype viruses were produced through the co-transfection of pCW57.1-*TP53*, pCAG-HIVgp, and pCMV-VSV-G-RSV-Rev into 293T cells using Lipofectamine 2000 (Thermo Fisher Scientific, Waltham, MA, USA). KMS26, OPM2, and KMS11 expressing pCW57.1-*TP53* (KMS26/Tet-on p53, OPM2/Tet-on p53, and KMS11/Tet-on p53, respectively) were obtained through infection with the pseudotype virus. Infected myeloma cells were selected using 0.5–1 µg/mL puromycin (Sigma-Aldrich, St Louis, MO, USA) and then cloned.

The three cell lines were cultured with 1 µg/mL doxycycline (TaKaRa Bio, Kyoto, Japan). Gene expression after 24 h of treatment was determined using real-time PCR. Cell proliferation was determined using WST-8 assays.

### 2.5. Treatment with Myc Inhibitor

MM.1S, KMS27, KMS28BM, KMS26, and OPM2 cells were treated with 20 or 50 µM Myc inhibitor 10058-F4 (Abcam, Cambridge, UK). Real-time PCR was used to determine gene expression after 48 h (MM.1S) or 72 h (remaining HMCLs) of treatment. Cell proliferation was determined using WST-8 assays.

### 2.6. MYC Activation in MYC-ER Cell Lines

Plasmid pBabepuro-myc-ER (#19128) was purchased from Addgene (Cambridge, MA, USA). U2OS, KMS27, and KMS28BM expressing MYC-ER (U2OS/MYC-ER, KMS27/MYC-ER, and KMS28BM/MYC-ER, respectively) were obtained through infection with the pBabepuro-myc-ER recombinant retroviral vector. Infected cells were selected using 1 µg/mL puromycin (Sigma-Aldrich, St Louis, MO, USA) and then cloned [17,18]. U2OS/MYC-ER, KMS27/MYC-ER, and KMS28BM/MYC-ER were cultured with 1 µM 4OHT tamoxifen (Sigma-Aldrich, St Louis, MO, USA). Gene expression after 96 h (U2OS/MYC-ER), 72 h (KMS28BM/MYC-ER), or 6 h (KMS27/MYC-ER) of treatment was determined using real-time PCR.

### 2.7. Isolation of Nucleic Acids

Total RNA, including miRNA, was extracted from the myeloma cell lines using a mirVana miRNA Isolation Kit (Ambion, Austin, TX, USA). RNA quantity and quality were measured using BioSpec-nano (SHIMADZU, Kyoto, Japan). Complementary DNA (cDNA) was synthesized using a PrimeScriptTM RT Reagent Kit with gDNA Eraser (TaKaRa Bio, Kyoto, Japan). MicroRNA cDNA was produced using a TaqMan^TM^ MicroRNA Reverse Transcription Kit (Thermo Fisher Scientific, Tokyo, Japan).

### 2.8. Real-Time PCR

Primary miRNA (pri-miRNA) and miRNA expressions were determined using real-time PCR with the TaqMan Gene Expression Master Mix (Applied Biosystems, Foster City, CA, USA). TP53 and MYC mRNA were determined as cDNA via real-time PCR using the Power^®^ SYBR Green PCR Master Mix (Thermo Fisher Scientific, Waltham, MA, USA). Primer sequences were as follows:

*TP53*: Forward, 5′-TCAGCATCTTATCCGAGTGGAA-3′; Reverse, 5′-TGTAGTGGATGGTGGTACAGTCA-3′; *MYC* Forward, 5′-CCTGGTGCTCCATGAGGAGA-3′; Reverse, 5′-CAGTGGGCTGTGAGGAGGTTT-3′; actin-β: Forward, 5′-TGGCACCCAGCACAATGAA-3′; Reverse, 5′-CTAAGTCATAGTCCGCCTAGAAGCA-3′.

### 2.9. Western Blot

The protein expression level of p53 was determined using Western blot. Cells were lysed in SDS-Lysis buffer (62.5 mM Tris pH6.8, 2% sodium dodecyl sulfate (SDS), 10% glycerol). The cell lysates were sonicated, and then the protein concentration of each sample was determined. β-mercaptoethanol and Bromophenol Blue were both added to the lysates at concentrations of 5%. Then, the lysates were boiled for 5 min and used as whole-cell lysates (WCLs). Equal amounts of protein (4–20 μg) were subjected to electrophoresis using a 12% precast polyacrylamide gradient gel (Bio-Rad Laboratories, Hercules, CA, USA). The electrophoretically separated proteins were transferred to a PVDF membrane (Immobilon-P Transfer Membrane, Merck KGaA, Darmstadt, Germany) using a semi-dry transfer apparatus (AE-6688 HorizeBLOT 4M, ATTO, Tokyo, Japan). The Western blot was performed using standard procedures. The Western blot used the following antibodies: anti-p53 (Cell Signaling Technology; p53 (DO-1) Mouse mAb #18032), anti-ACTB (Cell Signaling Technology; β-Actin (D6A8) Rabbit mAb #8457), Anti-mouse IgG (Cell Signaling Technology; Anti-mouse IgG, HRP-linked Antibody #7076), and Anti-rabbit IgG (Cell Signaling Technology; Anti-rabbit IgG, HRP-linked Antibody #7074).

### 2.10. Apoptosis Analysis

The myeloma cell lines were collected after treatment with the agents, washed twice with cold-PBS, and washed once with 7-AAD binding buffer (10 mM HEPES pH7.4, 140 mM NaCl, 2.5 mM CaCl_2_). After centrifugation and aspiration, 5 µL 7-AAD (BioLegend, San Diego, CA, USA) and/or 5 µL Annexin V conjugated to FITC (BioLegend, San Diego, CA, USA) was added to the suspension. The antibodies were incubated for 30 min at RT and then washed once in PBS. The suspension after centrifugation and aspiration was analyzed on a BD FACSCanto^TM^ Ⅱ flow cytometer (BD Biosciences, Franklin Lakes, NJ, USA).

### 2.11. Statistical Analysis

All statistical analyses were performed in EZR version 1.54 (Saitama, Japan) [19]. Significance was set at *p* < 0.05. The RT-qPCR data were analyzed using Student’s *t* test or the Mann–Whitney *U* test. Correlations were evaluated using Spearman’s rank method.

## 3. Results

### 3.1. miR-34 Family and TP53 mRNA Expressions and Their Correlations in the Patients

Mature miR-34a and 34b and *TP53* mRNA expressions in the bone marrow plasma cells were determined using RQ-PCR. Consistent with our previous data, both mature miR-34a and 34b expressions were lower in the MM plasma cells than in the MGUS plasma cells (*p* = 0.0063 and *p* < 0.001, respectively) (Figure 1A,B). However, *TP53* mRNA expressions were higher in MM than in MGUS (*p* = 0.0028) (Figure 1C). Contrary to the expectation, in the plasma cells obtained from the MM and MGUS bone marrow specimens, a negative correlation between *TP53* and mature miR-34a and 34b was observed (miR-34a: r = −0.402, *p* < 0.001, miR-34b: r = −0.341, *p* < 0.001) (Figure 1D,E). These results imply that miR-34 expression is not upregulated by *TP53*.

### 3.2. p53 Protein Accumulation and p53 Overexpression Upregulated Primary and Mature miR-34 in Human Multiple Myeloma Cell Lines (HMCLs)

P53 protein accumulation in response to nutlin-3 treatment (1 µM) significantly upregulated primary (pri-)miR-34a in the MM.1S cells with wild-type (WT) *TP53.* Mature miR-34a and miR-34b levels tended to increase (Figure 2A). Nutlin-3 (10 µM) did not significantly increase the pri-miR-34a expressions in KMS27, KMS28BM, KMS26, and OPM2 harboring mutant *TP53*, as well as in *TP53*-deficient KMS11, and nutlin-3 (10 µM) did not increase either primary or mature miR-34a/b (Figure 2B–F). The Western blot analysis showed increased p53 protein levels in MM.1S and KMS27, but the levels remained unchanged in KMS28BM, KMS26, and OPM2 after the treatment with nutlin-3. p53 protein was not detected in KMS11 (Figure 2G and Appendix A).

We then introduced Tet-on WT-*TP53* into KMS26, OPM2, and KMS11. Forced WT *TP53* overexpression significantly increased both pri-miR-34a and mature miR-34a/b in all three lines (Figure 3A–C).

### 3.3. MYC mRNA Expression in Patients

As described above, forced *TP53* expression can induce miR-34 expression. Thus, the *TP53*-miR-34 pathway may be inhibited by factors that are highly expressed/highly active in the patient specimen. *MYC* is a known driver oncogene for MM progression and a repressor of several microRNAs, such as the miR-29 family. So, we next examined *MYC* expressions in MM and MGUS. As expected, *MYC* expression was significantly higher in MM than in MGUS (*p* = 0.001) (Figure 4A). A weak negative correlation between *MYC* and miR-34a and 34b was observed (miR-34a: r = −0.30, *p* < 0.001; miR-34b: r = −0.19, *p* = 0.015) (Figure 4B,C).

### 3.4. Myc Inhibitor Alone Did Not Change miR-34 Family Expression in Most HMCLs

Because *MYC* suppresses miR-29 family expression, we attempted to clarify whether *MYC* is involved in regulating miR-34 family expression. Human MCLs were cultured with the Myc inhibitor 10058-F4, which attenuates MYC transcriptional function through dissociating the MYC-MAX transcription factor complex. The Myc inhibitor alone did not significantly increase pri-miR-34a or mature miR-34a in KMS27, KMS28BM, and KMS26, except for in OPM2 cells (Figure 5A–E).

### 3.5. WT p53 Accumulation and Myc Inhibition Synergistically Upregulated miR-34 Expression

Because p53 induces the miR-34 family, we investigated whether p53 accumulation and simultaneous Myc inhibition would have a synergistic effect on miR-34 expression. 

In the MM.1S cells, Myc inhibition enhanced the upregulation of pri-miR-34a and mature miR-34a induced by p53 accumulation from nutlin-3 treatment (Figure 6A). Myc inhibition also increased pri-miR-34a expressions in KMS27 and KMS28BM cells following nutlin-3 treatment (Figure 6B,C). The treatment of the MM.1S, KMS27, and KMS28BM cells with (-)-Nutlin-3 and the Myc inhibitor 10058-F4 began at the same time.

In the KMS26 and OPM2 cells with the Tet-on *TP53* system, Myc inhibition further increased pri-miR-34a expression due to forced p53 overexpression. Mature miR-34a expression also markedly increased in the KMS26 cells, indicating a synergistic effect between p53 accumulation and Myc inhibition (Figure 7A,B). The treatment of the Tet-on p53 KMS26 and OPM2 cells with doxycycline and the Myc inhibitor began at the same time.

### 3.6. Forced MYC Activation Repressed p53-Mediated miR-34 Expression in MYC-ER Cell Lines

Because Myc inhibition enhances p53-mediated miR-34 family expression, we examined whether MYC activation could suppress p53-induced miR-34 family expression. We used an MYC-ER cell line that expresses MYC but masks its nuclear translocation signal with the estrogen receptor ligand binding domain (ER). When tamoxifen (4OHT) binds to ER, the nuclear translocation signal is unmasked, allowing Myc translocation to the nucleus and increased activity [17,18]. We simultaneously induced p53 accumulation and MYC activation in three MYC-ER cell lines, namely, osteosarcoma cell line (1) U2OS harboring WT-*TP53*, and HMCLs (2) KMS27 and (3) KMS28BM.

In the U2OS/MYC-ER cells, MYC activation suppressed the p53-induced increase in mature miR-34a expression (Figure 8A). In the KMS27 cells, MYC suppressed the p53-induced upregulation of pri-miR-34a expression (Figure 8B). Finally, the KMS28BM cells exhibited the same changes in pri-miR-34a and mature miR-34a expressions as the U2OS/MYC-ER and KMS27 cells (Figure 8C).

### 3.7. MM Cell Proliferation and Apoptosis after Co-Treatment with Nutlin-3 and Myc Inhibitor

Because Myc inhibition enhances *TP53*-mediated miR-34 family expression, we next examined whether p53 accumulation and Myc inhibition synergistically altered MM cell proliferation and apoptosis. In MM.1S, nutlin-3 markedly reduced cell proliferation. The Myc inhibitor alone did not affect the proliferation and did not show synergistic effects. (Figure 9A) In KMS27 and KMS28BM, neither nutlin-3 nor the Myc inhibitor alone altered the proliferation, but the combination slightly suppressed the proliferation (Figure 9B,C). Apoptosis or cell death was not significantly increased by p53 accumulation or Myc inhibition ((Figure 9A–C and Appendix A).

Forced p53 expression markedly reduced the proliferation of Tet-on p53 KMS26 and Tet-on OPM2, but the Myc inhibitor, either alone or in combination with p53 expression, did not alter the proliferation (Figure 10A,B). Apoptosis or cell death was increased by forced p53 expression but was not increased by the Myc inhibitor in both cell lines.

## 4. Discussion

In this study, we demonstrated that p53 accumulation induced miR-34 family expression, an effect that was enhanced by inhibiting MYC activity. In contrast, forced MYC activation via the MYC-ER system suppressed TP53 to mediate miR-34 family expression in the MM cells. The MYC expression levels were inversely correlated with the mature miR-34 expression levels in the bone marrow plasma cells of MM and MGUS.

We found that nutlin-3, a drug that accumulates the p53 protein, increased miR-34a/b and pri-miR-34a in MM.1S with WT p53 and in some p53-mutant HMCLs but not in p53-deletion HMCLs. The miR-34 family is known to be induced by p53 and by genes related to cell growth arrest and apoptosis, such as p21, MDM2, and PUMA [15]. Furthermore, nutlin-3 induces the expressions of miR-192, 194, 215 [20], p21, MDM2, and PUMA [21] in p53-WT HMCLs. However, the drug’s capacity to induce miR-34 has not yet been exhibited in MM. Our current results are consistent with those of previous reports that nutlin-3 increases miR-34 expression in retinoblastoma and dopaminergic neuroblastoma cells [22,23]. The forced expression of WT p53 using the Tet-on system also induced mature miR-34a/b and pri-miR-34a expressions in all tested cell lines, indicating that WT p53 has the capacity to induce miR-34 family expression in MM.

However, we found that MM had a higher p53 expression than MGUS but a lower miR-34 family expression, suggesting a mismatch between TP53 and miR-34 in this cancer. Although frequently found in many cancer cells [24,25], deleterious TP53 mutations are rare in MM [21,22,24], meaning that TP53 dysfunction is unlikely to be causing miR-34 suppression. This discrepancy suggests the presence of factors that inhibit the p53 induction of miR-34 expression.

Many cancers exhibit abnormal MYC expressions [26], including MM [27]. Multiple studies have demonstrated that MYC functions as a transcriptional inhibitor of tumor suppressor genes [26], such as miR-34, miR-26, miR-15/-16, miR-23, miR-29, Let-7, and miR-126* [16,28,29,30,31]. Here, we found that an Myc inhibitor, which blocks heterodimer formation with MYC-associated factor X (MAX) [32], induced miR-34a and pri-miR-34a expressions in a concentration-dependent manner. However, this effect only occurred in OPM2 and not in the other HMCLs. This result indicates that MYC inhibition alone is insufficient to induce miR-34 family expression in most of the HMCL proliferative states.

Next, we simultaneously inhibited MYC while increasing miR-34 family expression by activating WT p53 with the nutlin-3 or Tet-on system. Although each HMCL had variable responses, MYC inhibition generally further increased miR-34a expression, suggesting that activated MYC represses p53-inducible miR-34.

Therefore, we investigated whether MYC suppresses p53-induced miR-34 family expression using MYC-ER cell lines that can be forced to activate MYC. In the WT TP53 osteosarcoma cell line (U2OS), MYC activation suppressed p53-induced miR-34a expression. For the two HMCLs, MYC activation reduced pri-miR-34a expression in the KMS27 cells, as well as pri-miR-34a and mature miR-34a expressions in the KMS28BM cells. Taken together, our findings support the hypothesis that MYC represses p53-induced miR-34 family expression.

Despite our results, there are still several unexplained phenomena. First, in diffuse large B-cell lymphoma, miR-34a expression is regulated in three ways: direct MYC mediation, epigenetic repression of the miR-34a promoter region, and miR-34a deletion [33]. However, the exact method of miR-34a regulation in MM is less clear. We successfully demonstrated a connection between MYC and miR-34 in plasma cells from the bone marrow of patients with MM and MGUS, specifically showing that TP53 and MYC were negatively correlated with mature miR-34 expression. Thus, MYC appears to suppress p53-induced miR-34 expressions in samples from patients and not only in the HMCL model.

We plan to perform studies using methods such as MYC ChIP assays to better understand the relationship between MYC and miR-34 in MM, clarifying whether MYC transcriptionally represses miR-34 family expression or acts via other mechanisms. While the in vitro results demonstrating this mechanism are convincing, further in vivo experiments need to be performed to demonstrate that this effect occurs in the presence of other microenvironmental mediators.

Although our results show that miR-34 expression was synergistically upregulated by p53 accumulation and Myc inhibition, the miR-34 upregulation was not translated to the suppression of proliferation or the cell death of HMCL. We used a relatively small amount of the Myc inhibitor because a larger amount greatly decreased miR-34 expression; thus, Myc inhibition did not affect cell proliferation or death. Our results might indicate that miR-34 plays roles other than affecting cell growth and survival.

In conclusion, we found that elevated WT p53 induced miR-34 family expression in MM cells, while elevated MYC suppressed miR-34 family expression. These patterns indicate that activated MYC can lead to MM development and malignant transformation because it inhibits p53-dependent miRNA expression, which functions as a tumor suppressor.

## Figures and Tables

**Figure 1 genes-14-00100-f001:**
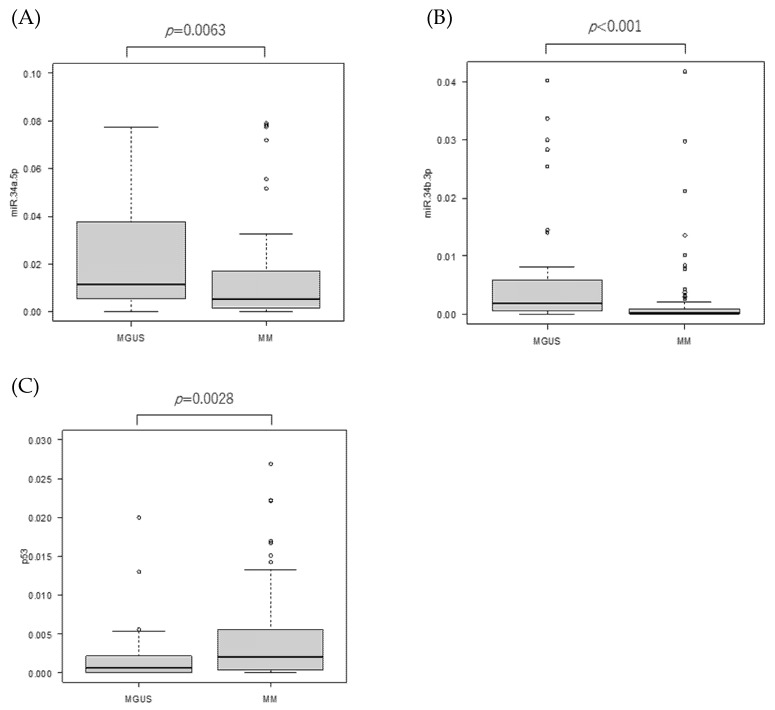
MiR-34 family and *TP53* mRNA expressions. Box plots show each transcript’s expression. Y axis indicates delta-delta Ct value. Outliers are excluded from the plots. (**A**) miR-34a, (**B**) miR-34b, (**C**) *TP53*. Correlations between miR-34 and *TP53* expressions. (**D**) miR34a-*TP53*, (**E**) miR-34b-*TP53*. Each dot represents an individual patient.

**Figure 2 genes-14-00100-f002:**
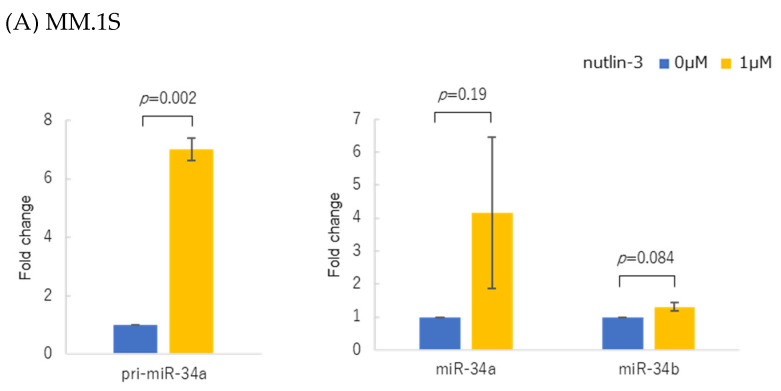
Changes in primary miR-34a (pri-miR-34a) and mature miR-34a/b expressions in response to MDM2 inhibitor nutlin-3 in six myeloma cell lines. (**A**) MM.1S was treated for 48 h with nutlin-3 at concentrations of 0 µM and 1 µM. (**B**) KMS27, (**C**) KMS28BM, (**D**) KMS26, (**E**) OPM2, and (**F**) KMS11 were treated for 72 h with nutlin-3 at 0 µM and 10 µM. Experiments were performed in triplicate. Error bars show the standard deviation (SD) across three experiments. Blue, with 0 µM nutlin-3; yellow, with 1 or 10 µM nutlin-3. (**G**) Western blot of p53 protein expression in response to nutlin-3 in myeloma cell lines. MM.1S was treated for 48 h with nutlin-3 at concentrations of 0 µM and 1 µM. KMS27, KMS28BM, KMS26, OPM2, and KMS11 were treated for 72 h with nutlin-3 at 0 µM and 10 µM. ND: Not detected.

**Figure 3 genes-14-00100-f003:**
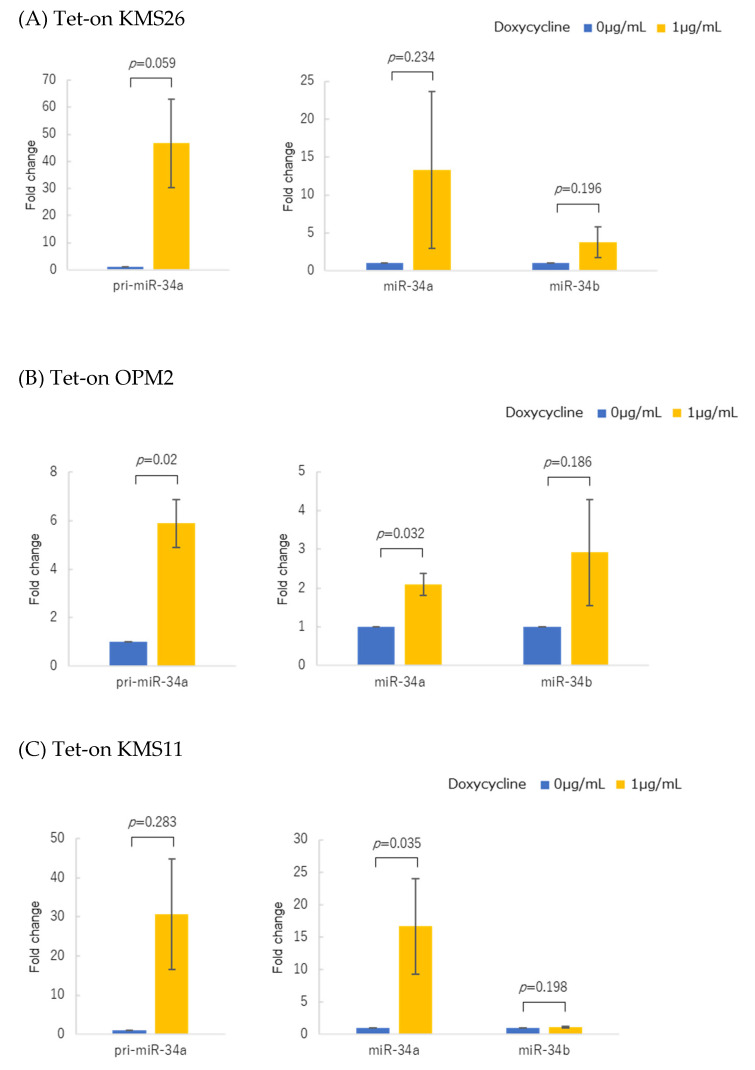
Changes in pri-miR-34a and mature miR-34a/b expressions in response to doxycycline in three Tet-on p53 myeloma cell lines. (**A**) KMS26, (**B**) OPM2, and (**C**) KMS11 were treated for 24 h with either 0 µg/mL or 1 µg/mL doxycycline. Experiments with KMS11 were performed four times, while the others were performed in triplicate. Error bars show the standard deviation (SD) across the experiments. Blue, with 0 µg/mL doxycycline; yellow, with 1 µg/mL doxycycline.

**Figure 4 genes-14-00100-f004:**
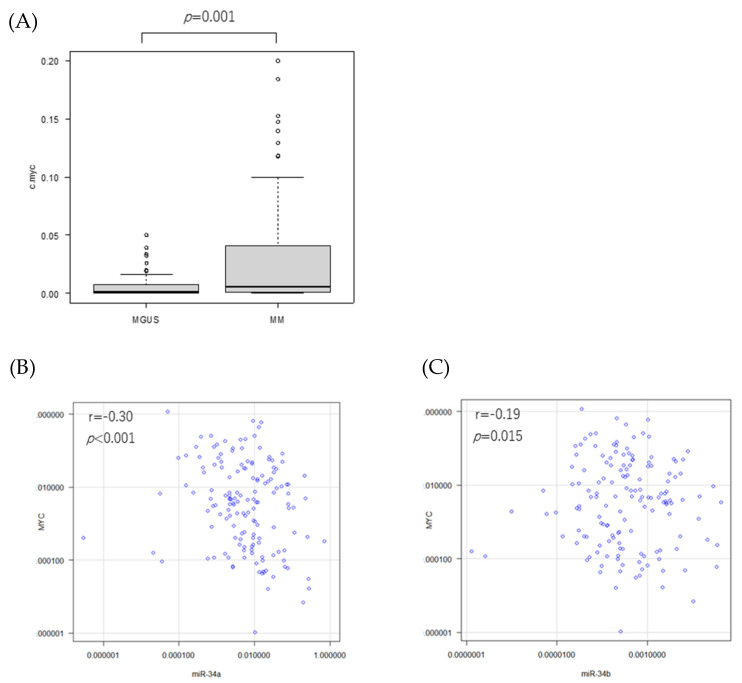
*MYC* and miR-34 expressions in the plasma cells obtained from the patients. (**A**) *MYC* expressions in MGUS and MM. Correlations between *MYC* mRNA and the miR-34 family for all patients. (**B**) *MYC* and mature miR-34a, (**C**) *MYC* and mature miR-34b. Each dot represents an individual patient.

**Figure 5 genes-14-00100-f005:**
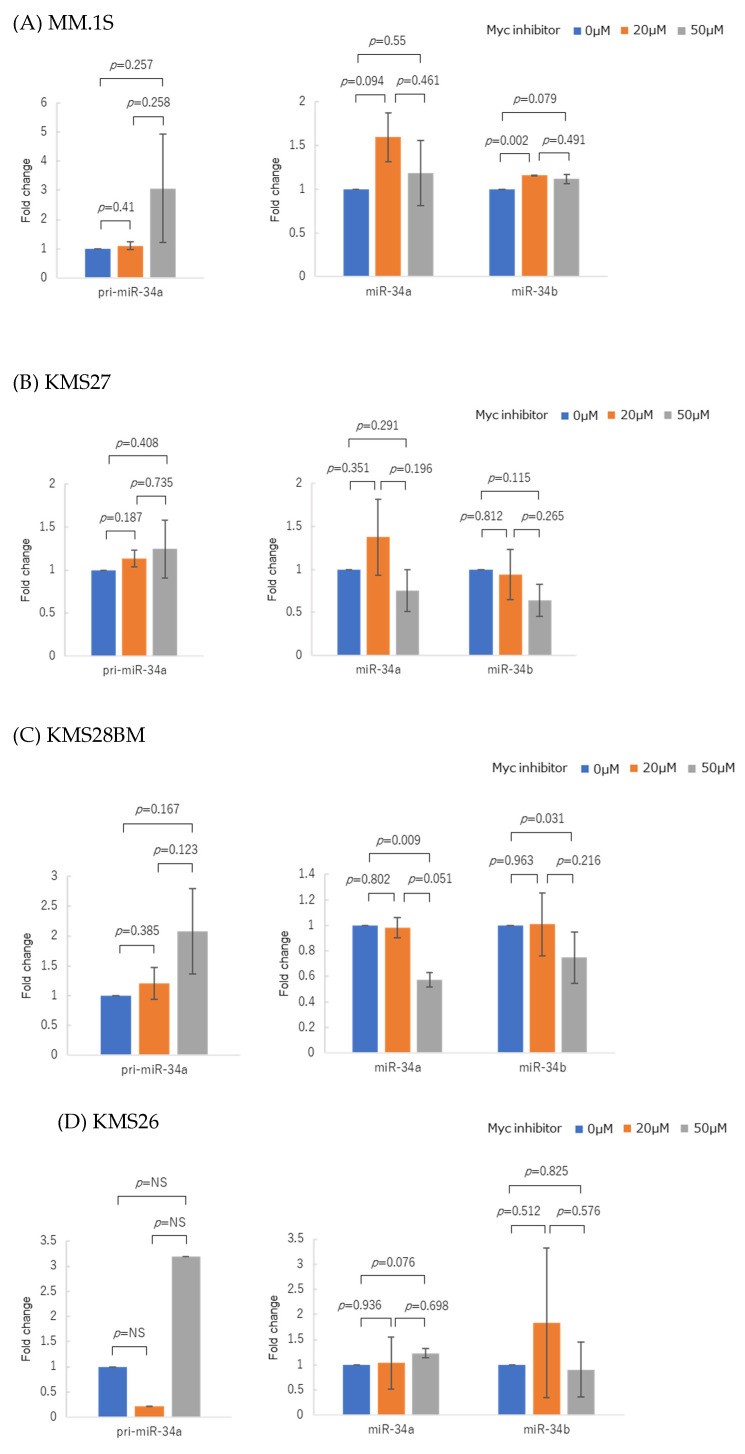
Changes in pri-miR-34a and mature miR-34a/b expressions in myeloma cell lines treated with Myc inhibitor 10058-F4. (**A**) MM.1S was treated for 48 h with Myc inhibitor at concentrations of 0 µM, 20 µM, and 50 µM. (**B**) KMS27, (**C**) KMS28BM, (**D**) KMS26, and (**E**) OPM2 were treated for 72 h with Myc inhibitor. Experiments were performed three times. Error bars show the standard deviation (SD) within triplicated experiments. Blue, with 0 µM Myc inhibitor; orange, with 20 µM Myc inhibitor; gray, with 50 µM Myc inhibitor. NS: Not significant.

**Figure 6 genes-14-00100-f006:**
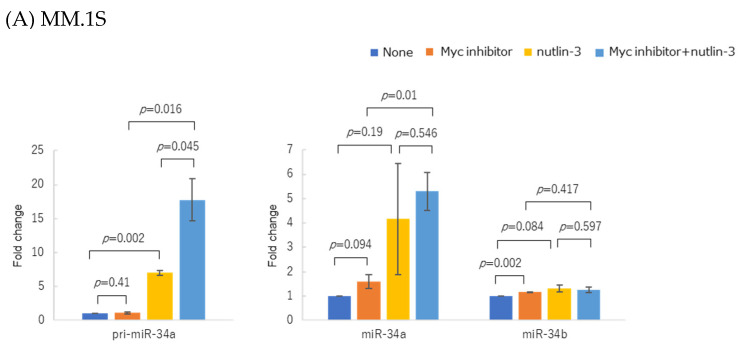
Changes in pri-miR-34a and mature miR-34a/b expressions in three myeloma cell lines after treatment with Myc inhibitor 10058-F4 and nutlin-3. (**A**) MM.1S, (**B**) KMS27, and (**C**) KMS28BM were treated with 20 µM of Myc inhibitor. MM.1S was treated for 48 h with 1 µM nutlin-3, while the other two cell lines were treated for 72 h with 10 µM nutlin-3. Experiments were performed in triplicate. Error bars show the standard deviation (SD) within triplicated experiments. Blue, no treatment; orange, with 20 µM of Myc inhibitor; yellow, with 1 or 10 µM of nutlin-3; light blue, with co-treatment of Myc inhibitor and nutlin-3. Blue and yellow bars show the same data as in Figure 2. Blue and orange bars show the same data as in Figure 5.

**Figure 7 genes-14-00100-f007:**
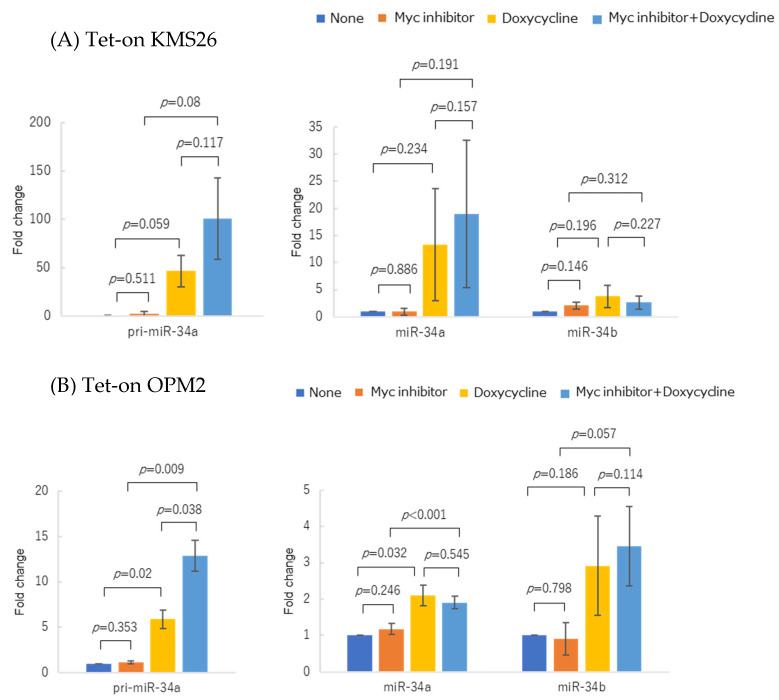
Changes in pri-miR-34a and mature miR-34a/b expressions in two Tet-on p53 myeloma cell lines after treatment with Myc inhibitor 10058-F4 and doxycycline. (**A**) KMS26 and (**B**) OPM2 were treated for 72 h with 20 µM Myc inhibitor and/or 1 µg/mL doxycycline. Experiments were per-formed in triplicate. Error bars show the standard deviation (SD) within triplicated experiments. Blue, no treatment; orange, with 20 µM Myc inhibitor; yellow, with 1 µg/mL doxycycline; light blue, with co-treatment of Myc inhibitor and doxycycline. Blue and yellow bars show the same data as Figure 3.

**Figure 8 genes-14-00100-f008:**
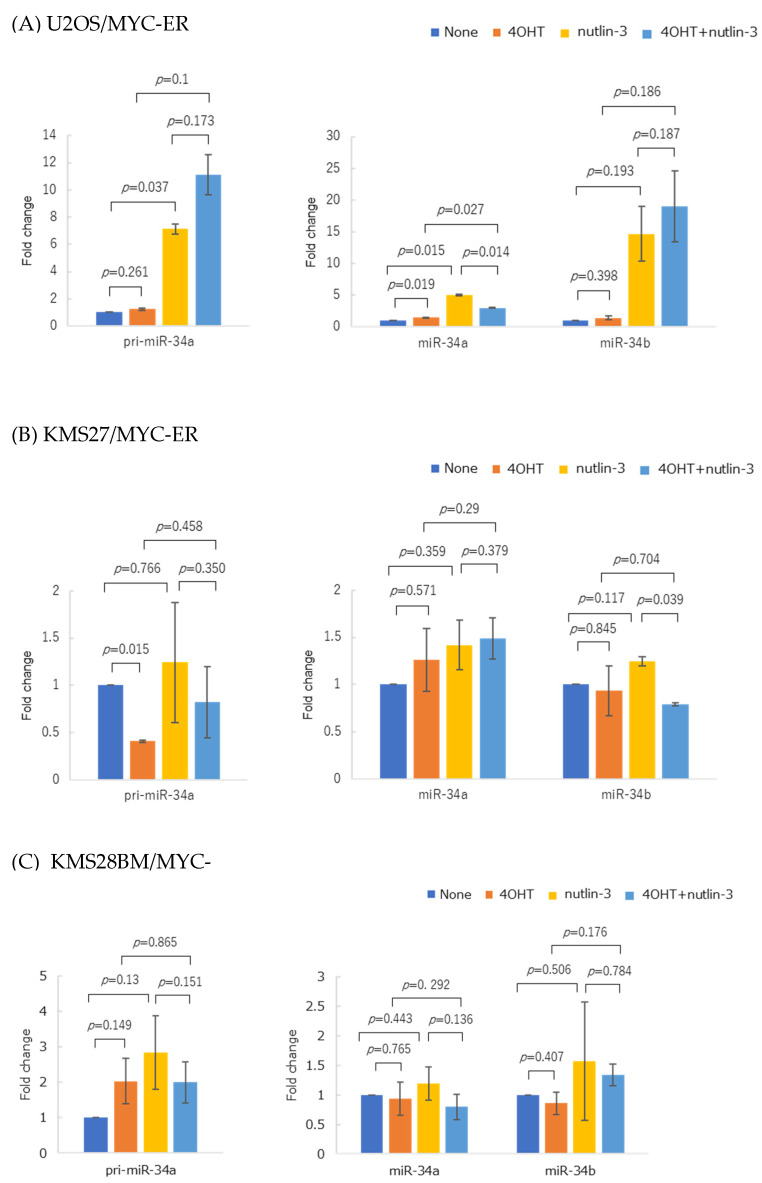
Changes in pri-miR-34a and mature miR-34a/b expressions in three MYC-ER cell lines after 4OHT tamoxifen and nutlin-3 treatment. (**A**) U2OS treated for 96 h with 1 µM 4OHT tamoxifen and 1 µM nutlin-3. (**B**) KMS27 treated for 6 h with 1 µM 4OHT tamoxifen and for 72 h with 10 µM nutlin-3. (**C**) KMS28BM treated for 72 h with 1 µM 4OHT tamoxifen and 10 µM nutlin-3. Only experiments with KMS28BM were performed in triplicate. Other experiments were performed twice. Error bars show the standard deviation (SD) across the experiments. Blue, no treatment; orange, with 1 µM 4OHT tamoxifen; yellow, with 1 or 10 µM nutlin-3; light blue, with co-treatment of 4OHT tamoxifen and nutlin-3.

**Figure 9 genes-14-00100-f009:**
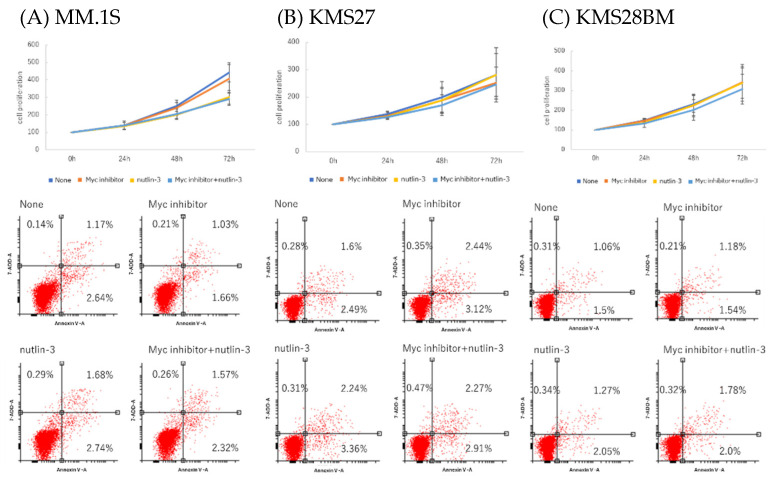
The myeloma cell line proliferation and apoptosis after treatment with Myc inhibitor 10058-F4 and nutlin-3. (**A**) MM.1S, (**B**) KMS27, and (**C**) KMS28BM were treated with 20 µM of Myc inhibitor. MM.1S was treated for 48 h with 1 µM nutlin-3, while the other two cell lines were treated for 72 h with 10 µM nutlin-3. Experiments of proliferation were performed in triplicate. Error bars show the standard deviation (SD) within triplicated experiments. Blue, no treatment; orange, with 20 µM of Myc inhibitor; yellow, with 1 or 10 µM of nutlin-3; light blue, with co-treatment of Myc inhibitor and nutlin-3. Apoptosis was measured once after treatment for 48 h or 72 h. In dot plot, left top, no treatment; right top, with 20 µM of Myc inhibitor; left bottom, with 1 or 10 µM of nutlin-3; right bottom, with co-treatment of Myc inhibitor and nutlin-3.

**Figure 10 genes-14-00100-f010:**
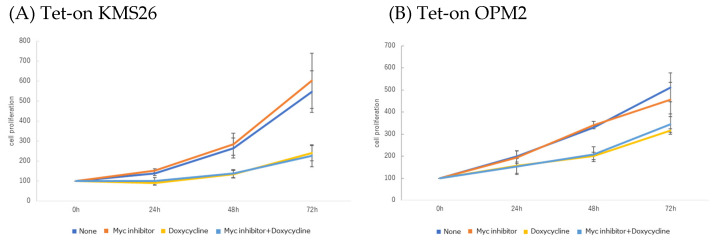
The Tet-on p53 myeloma cell line proliferation and apoptosis after treatment with Myc inhibitor 10058-F4 and doxycycline. (**A**) KMS26 and (**B**) OPM2 were treated for 72 h with 20 µM of Myc inhibitor and/or 1 µg/mL doxycycline. Experiments of proliferation were performed in triplicate. Error bars show the standard deviation (SD) within triplicated experiments. Blue, no treatment; orange, with 20 µM Myc inhibitor; yellow, with 1 µg/mL doxycycline; light blue, with co-treatment of Myc inhibitor and doxycycline. Apoptosis was measured once after treatment for 72 h. In dot plot, left top, no treatment; right top, with 20 µM of Myc inhibitor; left bottom, with 1 µg/mL doxycycline; right bottom, with co-treatment of Myc inhibitor and doxycycline.

**Table 1 genes-14-00100-t001:** Patients’ demographics. F: Female; M: Male; NP: not particular.

	Group	MGUS	MM
n		64	109
age		71 (38–88)	69.5 (44–88)
Gender (%)	F	37 (58.7)	53 (49.1)
	M	26 (41.3)	55 (50.9)
IgH (%)	BJ	2 (3.5)	19 (17.6)
	IgG	40 (70.2)	63 (58.3)
	IgA	11 (19.3)	22 (20.4)
	IgD	0 (0.0)	2 (1.9)
	IgM	2 (3.5)	0 (0.0)
	unknown	2 (3.5)	2 (1.9)
IgL (%)	κ	31 (54.4)	60 (55.6)
	λ	24 (42.1)	46 (42.6)
	unknown	2 (3.5)	2 (1.9)
ISS (%)	1	NA	22 (21.0)
	2	NA	45 (42.9)
	3	NA	38 (36.2)
R.ISS (%)	1	NA	12 (12.1)
	2	NA	72 (72.7)
	3	NA	15 (15.2)
Cytogenetics.Risk (%)	High	NA	35 (34.3)
	Standard	NA	67 (65.7)
Cytogenetics.Karyotype (%)	del 17p	NA	12 (12.5)
	t (11; 14)	NA	21 (21.9)
	t (14; 16)	NA	2 (2.1)
	t (4; 14)	NA	16 (16.7)
	trisomy11	NA	18 (18.8)
	NP	NA	27 (28.1)

## Data Availability

Not applicable.

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
