# Peer review of "MYC Causes Multiple Myeloma Progression via Attenuating TP53-Induced MicroRNA-34 Expression"

_genes, 2022, doi:10.3390/genes14010100_

Round 1
Reviewer 1 Report
The authors demonstrated that MYC suppressed wild type TP53-induced microRNA-34 expression in human myeloma cell lines. Although the precise mechanism is still unclear, the findings are novel and very interesting. However, it is difficult to interpret some results because TP53 status is different among myeloma cell lines used in the study (eg. wild type, mutated type, or deficient type). To avoid misunderstanding results, the authors should show TP53 protein expression status before and after treatment with nutlin-3 in myeloma cell lines tested.
Reviewer 2 Report
Here, Murakami et al. investigated the role of MYC in the regulation of the TP53-target miRNA, miR-34. They demonstrate that MM patients have lower levels of miR-34 but higher levels of TP53 mRNA and MYC that MGUS ones. Induction of TP53 by using the MDM-2 inhibitor nutlin-3 induces miR-34 expression; by contrast, inhibition of MYC did not increase miR-34 expression but the simultaneous inhibition of MYC and overexpression of TP53 was able to induce miR-34 expression suggesting that MYC is involved in the suppression of p53-dependent miR34 expression. This mecchanism may contribute to MM onset and malignant transformation.
The manuscript is interesting, innovative and methodologies are adequate. Some point need to be revised.
Mayor remarks:
- Authors demonstrate that the simultaneous inhibition of MYC and induction of p53 induce the overexpression of miR-34. What happens to MM cells survival and apoptosis after co-treatment and miR34 over-expression?
- I suggest to provide a table with the characteristic of MGUS and MM patients
Minor remarks:
- Author should specify in material and methods or in the main text whether the treatment with the MYC inhibitor, 10058-F4. and the MDM-2 inhibitor, nutlin-3, starts in the same moment or not.
- Author should specify the reason why most HMCLs were treated with nutlin-3 for 72 hours and MM.1S for 48 hours
Round 2
Reviewer 1 Report
The authors revised the manuscript appropriately according to reviewer's comment.